# Liquid Biopsy for Glioma Using Cell-Free DNA in Cerebrospinal Fluid

**DOI:** 10.3390/cancers16051009

**Published:** 2024-02-29

**Authors:** Ryosuke Otsuji, Yutaka Fujioka, Nobuhiro Hata, Daisuke Kuga, Ryusuke Hatae, Yuhei Sangatsuda, Akira Nakamizo, Masahiro Mizoguchi, Koji Yoshimoto

**Affiliations:** 1Department of Neurosurgery, Graduate School of Medical Sciences, Kyushu University, Fukuoka 812-8582, Japan; 2Department of Neurosurgery, Oita University Faculty of Medicine, Yufu 879-5593, Japan; 3Department of Neurosurgery, National Hospital Organization Kyushu Medical Center, Clinical Research Institute, Fukuoka 810-8563, Japan

**Keywords:** glioma, liquid biopsy, cell-free DNA, cerebrospinal fluid, digital PCR, next-generation sequencing

## Abstract

**Simple Summary:**

Glioma is one of the most common primary central nervous system (CNS) tumors, and its molecular diagnosis is crucial. However, surgical resection or biopsy is risky when the tumor is located deep in the brain or brainstem. Liquid biopsy, particularly targeting cell-free DNA (cfDNA) in cerebrospinal fluid (CSF), is a minimally invasive method, and emerges as a promising alternative, overcoming spatial and temporal heterogeneity in CNS tumors. Unlike extracranial cancers, cfDNA in the blood is scarce in CNS tumors, including glioma, emphasizing the relevance of CSF. Several successful studies have been conducted to detect tumor-specific genetic alterations in cfDNA from CSF using digital PCR and/or next-generation sequencing. This review provides an overview of the current status of CSF-based cfDNA-targeted liquid biopsy for gliomas. It delineates distinctions from liquid biopsies for extracranial cancers, addresses prevailing challenges, and explores future prospects in this field.

**Abstract:**

Glioma is one of the most common primary central nervous system (CNS) tumors, and its molecular diagnosis is crucial. However, surgical resection or biopsy is risky when the tumor is located deep in the brain or brainstem. In such cases, a minimally invasive approach to liquid biopsy is beneficial. Cell-free DNA (cfDNA), which directly reflects tumor-specific genetic changes, has attracted attention as a target for liquid biopsy, and blood-based cfDNA monitoring has been demonstrated for other extra-cranial cancers. However, it is still challenging to fully detect CNS tumors derived from cfDNA in the blood, including gliomas, because of the unique structure of the blood–brain barrier. Alternatively, cerebrospinal fluid (CSF) is an ideal source of cfDNA and is expected to contribute significantly to the liquid biopsy of gliomas. Several successful studies have been conducted to detect tumor-specific genetic alterations in cfDNA from CSF using digital PCR and/or next-generation sequencing. This review summarizes the current status of CSF-based cfDNA-targeted liquid biopsy for gliomas. It highlights how the approaches differ from liquid biopsies of other extra-cranial cancers and discusses the current issues and prospects.

## 1. Introduction

Glioma is one of the most common central nervous system (CNS) tumors, accounting for 80% of primary brain malignancies [1], and its molecular diagnosis is crucial [2]. To obtain a correct diagnosis and provide multidisciplinary treatment, surgical excision or biopsy followed by an integrated diagnosis is required, including molecular evaluation. Although stereotactic biopsy is a minimally invasive biopsy option, significant morbidity and serious complications can occur in 5–10% of cases, as reported in meta-analyses and population-level studies [3,4]. Even with biopsies, diagnosis is not possible in 3–17% of cases, and recent reports have shown no significant improvement in results [3,4,5,6]. In addition, in some cases, particularly in the deep brain or brainstem region, it is even riskier to perform a biopsy due to its location. In such cases, minimally invasive diagnostic options, such as liquid biopsy, may be beneficial in conjunction with surgery. Furthermore, recent studies have revealed diverse heterogeneous molecular landscapes in CNS tumors [7,8]; thus, there is a potential risk that tissue fragments obtained through biopsy reflect only partial clones of the entire tumor [9,10]. Liquid biopsy overcomes temporal and spatial heterogeneity and is an optimal solution to these problems [11]. In principle, the biomarkers in the collected specimens reflect all clones, even if they differ in proportion, which reduces sampling bias.

Although there are various potential targets for liquid biopsy, tumor-derived nucleic acids present in body fluids are considered promising because they directly reflect the molecular profile of the tumor itself. DNA released from cells into body fluids is called cell-free DNA (cfDNA), and tumor-derived cfDNA is called circulating tumor DNA (ctDNA) (see Section 2.1. for details). Indeed, ctDNA has been shown to reflect intratumor heterogeneity [12,13,14] and spatially separated foci in extracranial solid tumors [15,16]. This approach has also effectively identified mutations not captured in traditional biopsy samples, as evidenced by multiple studies [12,15,17]. Because of its minimally invasive nature, repeated longitudinal monitoring is practical, and it has already been revealed that ctDNA changes at the time of recurrence can be detected earlier than clinically apparent in breast, colorectal, and lung cancers [18,19,20,21,22]. These advantages of liquid biopsy are especially beneficial in gliomas with large heterogeneity within the tumors [7,23].

In clinical situations, the current clinical management of gliomas relies on diagnostic imaging, which does not have high sensitivity or specificity for the evaluation of recurrence after primary treatment [24,25,26,27,28,29]. For example, up to one-third of patients with GBM experience pseudoprogression, which is an abnormal contrast enhancement associated with improved prognosis; pseudoresponse, which is an apparent shrinkage of the contrast enhancement in patients treated with bevacizumab; and necrosis after radiation therapy, which cannot be accurately detected [25,26,27,28,29]. Effective liquid biopsy methods may solve these problems in the clinical management of glioma. Furthermore, advances in our understanding of the tumor microenvironment of glioma and advances in immunotherapy have underscored the importance of assessing bodily fluid biomarkers for therapeutic planning and clinical management [30,31,32,33].

Therefore, the establishment of liquid biopsy in gliomas is important for early diagnosis, detection of minimal residual disease after surgery, evaluation of disease status after treatment, identification of treatment resistance mechanisms, and prediction of outcome [34,35,36,37,38,39,40]. Among the various studies conducted to date, research that directly evaluates tumor-specific gene alterations using cfDNA has attracted remarkable attention. With the recent accumulation of molecular biological knowledge, the evaluation of characteristic genetic mutations is essential for the diagnosis of gliomas [2]; conversely, the demonstration of tumor-specific genetic mutations is the basis for the success of liquid biopsy. Unlike ctDNA obtained in cases of other extra-cranial cancer types, ctDNA obtained from the blood is extremely scarce in gliomas, making it reasonable to target ctDNA in the cerebrospinal fluid (CSF).

In this review, we summarize the current status of liquid biopsy of gliomas targeting ctDNA from CSF, including the differences from liquid biopsy for other cancer types, and discuss the current issues and prospects of liquid biopsy as a diagnostic technique. Specifically, Section 2 provides an overview of cfDNA/ctDNA-based liquid biopsy techniques and Section 3 summarizes existing methodologies. Section 4 summarizes the molecular background of various liquid biopsy targets for glioma and describes potential applications of these assays.

## 2. Background of Liquid Biopsy for Glioma and Cerebrospinal-Fluid-Derived Cell-Free DNA

### 2.1. Debates on Terminology: cfDNA vs. ctDNA

DNA in body fluids is released from cells via apoptosis or necrosis and exists as cfDNA, which are small fragments of nucleic acids that are not associated with cells or cell fragments [34,41,42,43]. cfDNA is derived from both tumor cells and normal tissues. Advances in genome analysis have made it possible to distinguish tumor-specific DNA from DNA derived from normal cells, such as peripheral leukocytes. This development has led to the use of the term ctDNA to describe cfDNA derived from tumor cells, which is distinct from cfDNA derived from normal cells (Figure 1).

The term “circulating” was introduced based on observations in peripheral blood-based liquid biopsy; however, even if the source is a body fluid other than blood, the term “circulating” tumor DNA is conventionally used [44,45,46]. Therefore, the term ctDNA is commonly used in CSF [44,45,47], although there is no free traffic with blood in the CSF (described in detail in Section 2.2), and CSF is not “circulating” [48,49]. In this context, some groups intentionally use the terms “tumor-derived cfDNA” or “cell-free tumor DNA”, using the same abbreviation, ctDNA [37,38,50]. The use of the term tumor-derived cfDNA in CSF is controversial.

In this review, we decided to use the term circulating tumor DNA (ctDNA), even when referring to DNA from the CSF, and we use cfDNA to describe CSF-derived DNA samples containing DNA from normal tissues.

### 2.2. Barriers to Liquid Biopsy of Gliomas: The Blood–brain barrier

Currently, methods to evaluate ctDNA in blood have been demonstrated in solid tumors, such as colorectal, breast, and lung cancers, and the detection of ctDNA in plasma as a companion diagnosis has been approved by the European Medicines Agency (EMA) and the United States Food and Drug Administration (FDA). Clinical trials using ctDNA as a guiding factor are currently ongoing (NCT02284633, NCT02743910, and ISRCTN16945804). More recently, the utilization of ctDNA for the molecular profiling of a wide range of solid tumor patients has become feasible, leading to early clinical trials [51]. These breakthroughs have been recognized as important milestones in the implementation of liquid biopsies in precision oncology [45]. However, in the shadow of this spectacular leap, liquid biopsy in the field of brain tumors, especially gliomas, continues to be a trial-and-error procedure. Compared to other cancer types, gliomas typically have a variant allele fraction (VAF) of less than 1% of ctDNA in plasma [52,53,54], and less than 10 copies of tumor-derived DNA in 5 mL of plasma [55]. This is believed to be due to the presence of the blood–brain barrier (BBB), a unique structure that characterizes the CNS and strongly restricts the entry of tumor-derived biomarkers into the bloodstream [52,55,56].

The BBB is composed of vascular endothelial cells, astrocyte ends, and pericytes, all of which are closely bound by tight junctions. This severely restricts the diffusion of substances between the blood and brain tissues [57,58,59,60,61]. Hydrophobic small molecules can pass through the BBB via lipid membranes; however, hydrophilic large molecules, including DNA, are assumed to be incapable of crossing the BBB [57,58,59,60,61]. Therefore, as long as the BBB functions normally, ctDNA cannot be released into the blood. However, there are situations in which the tight junctions of the BBB become dysfunctional owing to tumor invasion. This can be confirmed as a contrast–enhancement effect or brain edema around the tumor in clinical imaging findings [62,63,64,65]. For this reason, it has been reported that high-grade CNS tumors with contrast–enhancing foci and brain edema are more likely to release tumor-derived molecules into the blood [52,54,55,66,67,68,69,70,71,72]. Conversely, in low-grade gliomas, the BBB continues to function normally [65], making the capture of blood-derived ctDNA difficult [54,68,72,73]. The capture of ctDNA from the blood in gliomas is more challenging than in other cancer types, and there are certain technical limitations in targeting all gliomas, particularly low-grade gliomas.

Considering that brain tissue containing tumor cells is immersed in the CSF, it is reasonable to extract ctDNA from the CSF. In gliomas, the amount of ctDNA is several orders of magnitude higher in the CSF than in plasma or urine [13,53,54,56,73,74,75,76]. CSF is a good sample resource with high concentrations of ctDNA and low contamination of cfDNA derived from non-tumor cells such as peripheral blood cells. CSF is considered the gold standard source for the liquid biopsy of gliomas [13,53,54,56,73,74,75,76]. Although collecting CSF is somewhat more invasive than drawing blood, CSF is a feasible target for liquid biopsy because it can be collected easily at the bedside via a lumbar puncture.

### 2.3. Evaluation of Quantity and Quality of Cell-Free DNA

When glioma cells proliferate and die via apoptosis, necrosis, or immune response, tumor DNA is immediately expelled into the surrounding stroma and the CSF. During apoptosis, tumor chromosomal DNA is fragmented around nucleosome boundaries (140–180 bp) via endonucleases, resulting in a characteristic fragmentation pattern [41,77,78,79]. Some of these ctDNA fragments are thought to enter the blood via the disrupted BBB [80]. Although glioma ctDNA has been shown to be present in the CSF, plasma, and even urine [73,76], there is significantly less ctDNA in the blood of gliomas than in other solid tumors, as described above [52,53,54,55,56,73,75,76]. Therefore, the selection of cerebrospinal fluid as a resource to extract ctDNA is an important factor for successful liquid biopsy with higher sensitivity [13,53,54,56,73,74,75,76].

The amount of cfDNA recovered from CSF has been reported to be highly variable, suggesting that it is affected by tumor grade, high tumor burden, dissemination, tumor location relative to the CSF reservoir, and collection site (intracranial or lumbar puncture) [11,53,74,81,82,83,84]. Although the dynamics of cfDNA in CSF are not well understood, cfDNA is thought to have a short half-life (less than 2 h) [85,86], so it is assumed that CSF collected from the cerebral cistern, which is closer to the tumor, provides a more concentrated ctDNA than lumbar puncture. In fact, in the study of the CSF collection site and VAF in diffuse midline glioma (DMG), VAF was significantly higher in CSF collected closer to the tumor than CSF from more distant sites [56]. However, it has been reported that there is no significant difference in the detection rate of ctDNA between lumbar puncture and intracranial CSF collection [11]. Generally, the absolute amount of cfDNA is higher in glioma patients than in healthy patients because the turnover of tumor cells is more enhanced than that in normal tissues. Furthermore, the quantity of ctDNA is expected to decrease as tumor cell activity decreases with effective treatment, and conversely, to increase during refractory or recurrence. In fact, recent studies have longitudinally examined ctDNA in the plasma of GBM patients before and after therapeutic intervention and confirmed that ctDNA levels decrease in response to therapy and increase at relapse [72,87,88]. Quantification of cfDNA can be carried out quickly and at minimal cost using a spectrofluorometer. Therefore, the quantitative value of cfDNA can be utilized as the most primitive monitoring method, but its tumor-specificity is uncertain. Since the total amount of cfDNA may be affected by other factors such as inflammation, which are unrelated to the tumor burden, it is difficult to evaluate only cfDNA concentration as a biomarker [87,89]. Therefore, it is practical to use the cfDNA concentration as an adjunct to other markers.

Evaluating the quality of cfDNA, such as its fragment length, through electrophoresis might be beneficial [76,82,84,87,88]. As mentioned above, since cfDNA fragments at nucleosome boundaries, assessing the peak and distribution of DNA fragment sizes using electrophoresis may provide a basis for selecting suitable samples for subsequent analysis [84,88,90]. Furthermore, previous studies have highlighted that tumor-derived cfDNA is consistently shorter than normal-tissue-derived cfDNA [77,78]. Therefore, there is an opinion that using primers that target DNA less than 100 bp, which is rich in cfDNA; for example, designing primers for amplicons of approximately 60 bp, could be optimal [91]. In fact, it has been reported that ctDNA was detected using digital PCR with a 51 bp amplicon in 83.3% (five cases) of CSF samples of six promoters of telomerase reverse transcriptase (p*TERT*) mutant glioblastoma patients who did not respond to MLPA assays that generally target DNA of 100 bp or more [84], which may be helpful in the design and selection of assays. However, a study reported that there was no difference in detection sensitivity between assays with 88 and 113 amplicons [88]. In any case, the main principle seems to be targeting short amplicons while maintaining specificity, stability, and a low false positive rate. It may also be possible to enrich ctDNA by selecting fragments of shorter size from the entire cfDNA. For example, Mouliere et al. analyzed 13 CSF samples from patients with primary high-grade glioma and found enrichment of tumor DNA fragments of 90–150 base pairs using in vitro and in silico size selection methods [90]. They demonstrated that specific fragmentation signatures enriched in small fragments of 90–150 base pairs improved the detection of tumor DNA in CSF, with a median enrichment of more than two-fold in over 95% of cases and four-fold in over 10% of cases. This innovative approach may be an interesting alternative method for detecting ctDNA in CSF at an acceptable cost.

## 3. Methodology of Liquid Biopsy Using Cell-Free DNA

### 3.1. Digital PCR

Currently, digital PCR is the most frequently investigated and successful technique for liquid biopsy of gliomas [13,56,76,83,92,93,94]. Digital PCR is an endpoint PCR with less than one copy of the template distributed in each compartment and is a technique for the absolute quantification of the copy number of mutant/wild-type nucleic acids using Poisson distribution as the method of statistical analysis [95,96]. Although there are variations in digital PCR, mainly owing to differences in partitioning, droplet digital PCR (ddPCR), in which nucleic acids are encapsulated in droplets, is the most common method. Digital PCR can accurately calculate VAF from the ratio of mutant to wild-type droplets at a low cost and in a short time, and does not require complex bioinformatics skills for analysis. Notably, it is sensitive enough to detect rare mutant alleles, which are present in only 1 in 100,000 wild-types (the detection limit of VAF is approximately 0.001%) [97]. It can be tested with smaller amounts of template DNA and is the most sensitive and specific method compared to Sanger sequencing, qPCR, and next-generation sequencing (NGS) [56].

Digital PCR is ideal for the detection of known single-nucleotide variants (SNVs), and in gliomas, it is reasonable to apply this technique to search for driver mutations in genes such as IDH1/2, TERT promoter (pTERT), histone H3 (H3.3 H3F3A, H3.1 HIST1H3B, etc.), and BRAF. This highly sensitive technique successfully detected ctDNA targeting H3 K27M and pTERT mutations in gliomas [56,76,83,98,99]. Table 1 summarizes the representative studies utilizing digital PCR on CSF and shows that digital PCR can detect variant alleles with a high sensitivity of 70–100% in glioblastoma and 80–100% in DMG [13,56,76,83,92,93,100]. However, even with digital PCR using CSF, the detection of low-grade gliomas is challenging, as they are generally detected in less than 50% of cases, and in some cases, they are not identified at all.

Having shown that CSF cfDNA can be used to detect tumor-specific mutations, the next issue to consider is its application in longitudinal monitoring. In a study using an in vitro model in which H3F3A K27M-mutant DMG cells were co-cultured with normal human astrocytes, the presence of H3F3A K27M mutation in cfDNA in the culture medium was evaluated using ddPCR, and H3F3A K27M mutant ctDNA was shown to reflect tumor growth [101]. In the ONC201 trial (NCT03416530), Cantor et al. performed serial CSF collection through lumbar punctures in 17 patients with H3F3A K27M mutant DMG [93,102]. This prospective clinical trial demonstrated the feasibility of disease monitoring. CSF ctDNA was evaluated using ddPCR and the VAF of H3F3A K27M was recorded. The H3F3A K27M mutation was detected in 96.5% of CSF samples, a correlation between the change in VAF compared to baseline and PFS was shown, and PFS was significantly prolonged in cases with decreased VAF during the course. A VAF elevation of ≥25% occurred 1–3 months prior to progression in 5/11 cases (45.4%). Furthermore, the pattern of change in VAF over time was useful in differentiating pseudo-progression from actual progression and pseudoresponse after bevacizumab treatment from true response.

Thus, the detection of the gene mutation of ctDNA in CSF using ddPCR is a near-complete liquid biopsy methodology for patients with known driver mutations. This is an ideal method, especially for high-grade glioma and DMG in children, where the risk involved in biopsy surgery is high. However, ddPCR has limited multiplexing capability because it requires primers and probes specific for a predefined mutation or target locus. Therefore, the amount of specimens required increases in proportion to the number of assays, however, the limited number of clinical specimens makes repeated testing impossible. Therefore, as described in the next section, parallel testing methods such as NGS can be employed to evaluate multiple loci or copy number alterations at a wide range of loci.

**Table 1 cancers-16-01009-t001:** Representative studies for cell-free DNA in cerebrospinal fluid using digital PCR.

Authors (Year) [Citation]	N	Diagnosis *1 (n)	CSF Collection	Target	Result
De Mattos-Arruda et al. (2015) [13]	6 *2	GBM (3) metastases (3)	LP *3	*IDH1*, *TP53*, *ANK2*, *EGFR*, *PTEN*, *FTH1*, *OR51D1*	Assays were designed as selected by WES of tumor. GBM 100% (3/3) Metastases 100% (3/3)
Martínez-Ricarte et al. (2018) [92]	20	GBM IDH-wt (8) Gliosarcoma (1) GBM IDH-mut (1) AA (2) DA (1) OD (4) DMG (3)	LP (17) Postmortem (2) VP shunt (1)	*IDH1*, *IDH2*, *TP53*, *H3F3A*, *pTERT*	GBM IDH-wt 87.5% (7/8) Gliosarcoma 100% (1/1) GBM IDH-mut 100%(1/1) AA 50% (1/2) DA 0% (0/1) OD 50% (2/4) *4 DMG 100% (3/3)
Panditharatna et al. (2018) [56]	28 *5	DMG (28) [H3F3A K27M (21), HIST1H3B K27M (6), H3 wild-type (1)]	EVD (4) Surgical site (4) Postmortem (22)	*H3F3A*, *HIST1H3B*, *ACVR1*, *PIK3R1*, *BRAF*	Successfully detected H3 K27M mutations in 75% of CSF collected at diagnosis, 67% of CSF collected during treatment, and 90% of CSF collected at postmortem. Feasibility of detecting mutant obligate partners in *ACVR1*, *PIK3R1*, or *BRAF* was shown. *H3F3A* K27M 85.7% (18/21) *HIST1H3B* K27M 100% (6/6)
Izquierdo et al. (2021) [76]	9 *6	DMG, pHGG *6	N/A	*H3F3A*, *ACVR1*, *TP53*	Variant allele was detected in 66.7% (6/9). Detection was 60% (3/5) in *H3F3A* K27M cases, whilst single positive droplets were found in two negative cases.
Fujioka et al. (2021) [83]	34	GBM IDH-wt (7) *7 GBM IDH-mut (4) DMG (5) AA (2) DA (1) AO (8) OD (1) Other glioma (6) *8	LP (11) Surgical site (23)	*IDH1*, *H3F3A*, *pTERT*	GBM IDH-wt 71.4% (5/7) GBM IDH-mut 100% (4/4) DMG 80% (4/5) AA 50% (1/2) DA 0% (0/1) AO 100% (8/8) OD 0% (0/1) Other glioma 100% (6/6)
Cantor et al. (2022) [93]	17 *9	DMG (17)	LP	*H3F3A*	Successfully detected H3 K27M mutations in 96.5% (28/29) of CSF and 85.4% (53/62) plasma samples. ctDNA was used for longitudinal tumor monitoring. Correlation between the change in VAF compared to baseline and PFS was shown, and PFS was significantly prolonged in cases with decreased VAF during the course. VAF elevation of ≥25% occurred 1–3 months prior to progression in 5/11 cases (45.4%).
Orzan et al. (2023) [100]	Cohort 1: 45	GBM IDH-wt (42) Astrocytoma G4 (1) Oligodendroglioma G3 (1) others (1)	Surgical site (45)	*IDH1*, *TP53*, *PTEN*, *pTERT*, *CDKN2A*, *CDK4*, *PDGFRA*, *EGFR*	In 36 samples eligible for ddPCR, at least one genetic alteration (SNV or CNV) was found in 25 samples (69.4%). GBM IDH-wt 68.6% (24/35) Astrocytoma G4 100% (1/1)
	Cohort 2: 40 (39 patients)	GBM IDH-wt (32) Astrocytoma G4 (2) Astrocytoma G2/3 (2) Oligodendroglioma G2/3 (2) others (2)	LP (40)	*IDH1*, *pTERT*, *EGFR*, *CDKN2A*	In 38 eligible samples to ddPCR, ITEC (*IDH1-pTERT-EGFR-CDKN2A*) protocol was successfully performed in 26 cases (68.4%), and successfully diagnosed in 21 cases (55.3%). GBM IDH-wt 66.7% (20/32) Astrocytoma G4 50% (1/2) Astrocytoma G2/3 0% (0/2) Oligodendroglioma G2/3 0% (0/2)

*1 Tumor diagnoses followed those of the authors at the time of reporting, unless otherwise noted. Gliomas are highlighted in bold. *2 The total number of cases in the study using WES was 12. *3 Brain metastases of breast cancer were collected from the cisterna magna during warm autopsy. *4 *pTERT* was not detected in one case of OD. *5 There were 110 samples, from 48 patients. This included 30 CSF samples, 79 plasma samples, and one cystic fluid sample. *6 There were 43 samples from 32 patients. There were 27 plasma, 6 samples serum samples, and 1 cystic fluid sample. The breakdown of the number of cases for each diagnosis was unknown. *7 Two AA and two DA cases with IDH-wild-type and *pTERT* mutant were assigned to GBM. *8 *IDH*, *pTERT*, and *H3F3A* were all negative; GBM, AA, and DA were included. *9 A total of 29 CSF and 62 plasma samples were collected from 17 patients. AA, anaplastic astrocytoma; ACVR1, activin A receptor type 1; ANK2, ankyrin 2; AO, anaplastic oligodendroglioma; CDK4, cyclin dependent kinase 4; CDKN2A/B, cyclin dependent kinase inhibitor 2A/B; DA, diffuse astrocytoma; DMG, diffuse midline glioma; EGFR, epidermal growth factor receptor; EVD, external ventricular drainage; FTH1, ferritin heavy chain 1; GBM, glioblastoma; IDH1/2, isocitrate dehydrogenase 1/2; LP, lumber puncture; mut, mutant; N/A, not available; OR51D1, olfactory receptor family 51 subfamily D member 1; PDGFRA, platelet derived growth factor receptor alpha; pHGG, pediatric-type diffuse high-grade glioma H3-wild-type and IDH-wild-type; PIK3CA, phosphatidylinositol-4,5-bisphosphate 3-kinase catalytic subunit alpha; PIK3R1, phosphoinositide-3-kinase regulatory subunit 1; PTEN, phosphatase and tensin homolog; pTERT, promoter of telomerase reverse transcriptase; OD, oligodendroglioma;WES, whole-exome sequencing; wt, wild-type.

### 3.2. Next-Generation Sequencing (NGS)

Along with digital PCR, NGS is currently the most actively studied method and has accumulated results [11,13,53,73,75,81,92,100,103,104,105,106,107]. NGS is a massively parallel sequencing technology widely used for genetic evaluation. The “first generation” DNA sequencing using chain termination invented by Sanger et al. in 1977 was a major technological breakthrough at the time [108] and is still a robust method applicable to the sequencing of specific regions. For example, Parsons et al. sequenced 20,661 protein-coding genes and found *IDH1* mutations in a subset of glioblastomas (astrocytoma, IDH-mutant, Grade 4 in WHO 2021 classification), which became the cornerstone for the molecular diagnosis of gliomas [109]. However, first-generation sequencing has parallel multiplexing problems and a limited throughput. Therefore, a completely different “next-generation” high-throughput sequencing technology has been developed for broader genome analysis. In brief, NGS enables a series of reactions to be performed in a massively parallel manner, such as the creation of DNA libraries of short fragments, amplification, and sequencing, using base-by-base elongation reactions, and can generate a large amount of data in an overwhelmingly short time compared to conventional methods.

Applications of NGS include targeted gene panels [110,111], whole-exome sequencing (WES) [112,113,114,115], and whole-genome sequencing (WGS). Currently, a vast amount of mutation data on protein-coding regions related to cancer has been accumulated, and most driver genes found at high frequencies have been identified [115,116]. Furthermore, cancer genome analysis of tumor tissues can be extended to WGS, which covers the entire genome and can also detect variants in the genome structure, and may lead to the discovery of new and currently undiscovered oncogenic aberrations [117,118,119,120,121,122,123,124,125].

Targeted gene panels are designed to inexpensively and efficiently search for tumor-associated genes that have been narrowed down through the accumulation of tumor tissue analyses [110,111]. Such gene panels can evaluate up to several hundred genes with high sensitivity and accuracy in deep sequencing and have been applied to companion diagnostics to enable precision oncology in cancer patients. These gene panels are the most widely used methods for liquid biopsy [21,22].

One of the main challenges in adapting NGS-based assays to liquid biopsy is the low frequency of the variant alleles. In principle, NGS-based molecular diagnostics are limited in detecting variants with frequencies of 1% or less owing to the high error rate introduced by the sequencing process [126,127,128,129]. To overcome this, molecular barcoding and bioinformatics approaches have been developed to reduce background noise, making it possible to detect variants with frequencies lower than 0.1–0.5% VAF [128,130,131,132,133,134,135]. Leveraging these approaches, companion diagnosis using ctDNA has now been established for solid tumors, such as colorectal cancer, breast cancer, and lung cancer, as described above. Similarly, the application of targeted gene panels, benefiting from these approaches, is now being extended to studies in gliomas.

Highly sensitive NGS-based assays have been shown to enable the detection of ctDNA in plasma samples from patients with glioma [52,54,72,136,137]; however, in addition to biological and technical noise, genomic alterations known to be associated with cancer have been found in plasma, even in healthy individuals, which may confound the interpretation [134,137,138]. In gliomas, as previously mentioned, the use of plasma ctDNA has its limitations. Instead, the focus has shifted to CSF ctDNA because of its high quality as a biofluid with minimal non-tumor-derived cfDNA, enabling more accurate research [74,81]. Representative studies using NGS to target CSF-derived ctDNA are summarized in Table 2.

**Table 2 cancers-16-01009-t002:** Representative studies for cell-free DNA in cerebrospinal fluid using next-generation sequencing.

Authors (Year) [Citation]	N	Diagnosis *1 (n)	CSF Collection	Method (n)	Detected Altered Genes	Result
Wang et al. (2015) [81]	35	GBM (10) DMG (2) *2 AA (1) DA (2) Other gliomas (5) Others (15)	Surgical site	Targeted sequencing (13) WES (SafeSeqS) (22)	*TP53*, *IDH1*, *pTERT*, *NF2*, *PIK3R1*, *PTCH1*, *PTEN*	Tumor-specific alterations were detected in 70% (14/20) of gliomas. GBM 100% (10/10) DMG 100% (2/2) *2 AA100% (1/1) DA 0% (0/2) other gliomas 20% (1/5) others 80% (12/15)
De Mattos- Arruda et al. (2015) [13]	12	GBM (4) *3 Metastases (8)	LP *4	Targeted sequencing (MSK-IMPACT) WES (Nextra Rapid Capture Exome kit [37 Mb])	*TP53*, *pTERT*, *PIK3CG*, *EPHB1*	Tumor-specific alterations were detected in all cases. GBM 100% (4/4) *5 metastases 100% (8/8)
Pentsova et al. (2016) [103]	53	GBM (4) AA (1) AO (3) OD (1) Other gliomas (2) Metastases (32) Others (10)	LP (52) VAD(1)	Targeted sequencing (MSK-IMPACT)	*IDH1*, *EGFR*, *PTEN*, *1p/19q-codel*, *CDK4*, *PIK3CA*, *PDGFRA*, *CDKN2B*	Tumor-specific alterations were detected in 54.5% (6/11) of gliomas. GBM 75% (3/4) *6 AA 100% (1/1) *6 AO 33.3% (1/3) *6 OD 0% (0/1) other gliomas 50% (1/2) metastases 62.5% (20/32) others 0% (0/10)
Martínez-Ricarte et al. (2018) [92]	20	GBM IDH-wt (8) Gliosarcoma (1) GBM IDH-mut (1) AA (2) DA (1) OD (4) DMG (3)	LP (17) Postmortem (2) VP shunt (1)	Targeted sequencing (Custom panel, 4 genes)	*IDH1*, *IDH2*, *ATRX*, *TP53*	Tumor-specific alterations were detected in 20% (4/20) of gliomas. GBM IDH-wt 12.5% (1/8) Gliosarcoma 0% (0/1) GBM IDH-mut 100% (1/1) AA 100% (2/2) DA 0% 0% (0/1) OD 50% 0% (0/4) DMG 100% 0% (0/3)
Juratli et al. (2018) [75]	38	GBM (38) [IDH-wt and *pTERT* mut]	Surgical site	Targeted sequencing for *pTERT*	*pTERT*	*pTERT* mutations were detected in 92.1% (35/38) of gliomas. Correlation between *pTERT* VAF and OS was shown; the lower quartile or the lower third VAF had significantly longer OS compared with high VAF. Longitudinal CSF sampling showed postoperative *pTERT* mutation detection from CSF (LP) was related to shorter PFS.
Miller et al. (2019) [53]	85	GBM (46) LGG (39)	LP *7	Targeted sequencing (MSK-IMPACT)	*IDH1*, *pTERT*, *1p/19q-codel*, *TP53*, *CDKN2A/B*, *EGFR*, *CIC*, *ATRX*, *PTEN*, *NF1*, *PIK3CA*, *CDK4*, *PIK3R1*, *RB1*, *PDGFRA*	Tumor-specific alterations were detected in 49.4% (42/85) The presence of ctDNA in the CSF was associated with shorter survival. Patients who had ctDNA in their CSF experienced a four-fold higher risk of death than subjects who did not.
Pan et al. (2019) [103]	57	Brainstem glioma (57)	Surgical site (54) LP (3)	Targeted sequencing (Custom panel, 68 genes)	*H3F3A*, *TP53*, *ATRX*, *PDGFRA*, *FAT1*, *HIST1H3B*, *PPM1D*, *IDH1*, *NF1*, *PIK3CA*, *ACVR1*	At least one tumor-specific mutation was detected in 82.5% (47/57). Among cases in which tumor-specific alterations were detected in the primary tumor, alterations were matched in 97.3% (36/37) and all alterations were detected in 83.8% (31/37). Tumor-specific alterations were readily detected in the CSF-derived cfDNA in 30% (3/10) of cases in which alterations were undetected in the tumor DNA.
Zhao et al. (2020) [104]	17	GBM, IDH-wild-type (4) AA, IDH-wt (2) AA, IDH-mut (2) DA, IDH-mut (4) DA, IDH-wt (2) AO, IDH-mut and 1p/19q codel (3)	Surgical site	Ion Torrent Ampliseq Cancer Panel	*FGFR1/3*, *APC*, *EGFR*, *RB1*, *SMAD4*, *ERBB2*, *KDR*, *IDH1/2*	At least one mutation was detected in all 17 cases, and 88.2% (15/17) had mutations concorded with the tumor tissue.
Bale et al. (2021) [105]	148 samples (137 patients)	HGG (28) LGG (14) Metastases (54) Others (52)	N/A	Targeted sequencing (MSK-IMPACT)	*TP53*, *EGFR*, *pTERT*, *ATRX*, *PIK3CA*, *NF1*, *IDH1*	Tumor-specific alterations were detected in 50.7% (75/148) of the samples. HGG 67.9% (19/28) LGG 7.1%(1/14) Among cases in which tumor-specific alterations were detected in the primary tumor, alterations were matched in 93.6% (44/47). In all, of the 358 variants detected in CSF cfDNA samples with baseline tumor sequencing, 293 were also identified in the tissue.
Miller et al. (2022) [11]	64 samples (45 patients)	HGG (10) LrGG (4) Others (31)	LP or VAD (46) Surgical site (18) *8	Targeted sequencing (MSK-IMPACT)	*ATRX*, *TP53*, *H3F3A*, *PDGFRA*,*CDKN2A*, *MYC*, *NF1*	Somatic alterations were detected in 30/64 samples (46.9%) and in at least one sample per unique patient in 21/45 patients (46.6%). HGG 70% (7/10) LGG (0/4) Matched tumor/CSF pairs were analyzed to compare the mutational profiles, and the shared mutation rate was 32.1% (18/56) in HGG.
Pagès et al. (2022) [73]	67 samples (54 patients) *9	HGG/HGNT (10) LGG/LGNT (14) Others (43)	N/A	ULP-WGS/ Targeted sequencing (Custom panel, 40 genes)	*H3F3A*, *HIST1H3B*, *TP53*	Successfully ULP-WGS performed in 68.7% (46/67) of samples. HGG or HGNT 7/10 LGG or LGNT 10/14 Only 3 HGG (DMG) were positive for tumor fraction analyzed by ichorCNA. Ten CSF cfDNA samples were applied to the targeted sequencing (including 2 HGG and 2 LGG/LGNT). Only 5 cases were positive for gene alteration; 2 HGG (DMG) were detected as altered *H3F3A* and *HIST1H3B*, respectively.
Orzan et al. (2023) [100]	Cohort 1: 45	GBM, IDH-wt (42) Astrocytoma G4 (1) Oligodendroglioma G3 (1) Others (1)	Surgical site	Targeted sequencing (Custom panel, 54 genes)	*PTEN*, *IDH1*, *ATRX*, *TP53*, *ASCL1*, *RB1*, *PIK3R1*, *MSH6*, *EGFR*, *PDGFRA*, *AKT1*	Three cases were analyzed (2 GBM and 1 astrocytoma G4). SNVs and CNVs were concorded with tumor DNA.
	Cohort 2: 40 (39 patients)	GBM, IDH-wt (32) Astrocytoma G4 (2) Astrocytoma G3 (2) Oligodendroglioma G2/3 (2) Others (2)	LP		*IDH1*, *TP53*, *POLD1*, *CIC*, *CDKN2A/B*, *PTEN*, *MLH3*, *NOTCH1*, *CDK6*, *PDGFRA*, *MYC*, *PMS2*, *ATRX*	Comparative analysis between CSF and tumor DNA was performed in 5 cfDNA containing at least 10 ng of DNA, and tumor-specific alterations were detected in 4 samples. Unlike in Cohort 1, the overlapping degree with tumor DNA was only partial.

*1 Tumor diagnoses followed those of the authors at the time of reporting unless otherwise noted. Gliomas are indicated by bold type. *2 The two cases with H3K27M mutations in the mesencephalon were classified as DMG. However, neither *H3F3A*(0/1) nor *HIST1H3C*(0/1) was detected. *3 One of the cases was IDH-mutant, corresponding to Astrocytoma G4. *4 Brain metastases of breast cancer were collected from the cisterna magna during warm autopsy. *5 IDH mutation was not detected (0/1), *pTERT* mutation was detected in 1/3 of cases, and mutations in *TP53* and *PIK3CG* were consistent with tumors. *6 Amplification of *EGFR*, *PDGFRA*, and *CDK4* was detected in three cases of GBM; *IDH1* R132H was detected in both AA and AO, and 1p/19q co-deletion was confirmed in AO. *7 A total of 82 LPs were detected; three were obtained from surgical sites or VP shunts, and all were postoperative. All were postoperative: 99% after radiotherapy (84/85) and 95% after chemotherapy (81/85). *8 Surgical sites in 5 cases, Ommaya reservoir, or VP shunt implantation intraoperatively in 13 cases. *9 Overall, 562 samples were obtained from 258 patients. In addition to the CSF, 257 plasma samples from 243 patients and 240 urine samples from 224 patients were collected. AA, anaplastic astrocytoma; ACVR1, activin A receptor type 1; AO, anaplastic oligodendroglioma; ASCL1, achaete-scute family bHLH transcription factor 1; CDK4/6, cyclin dependent kinase 4/6; CDKN2A/B, cyclin dependent kinase inhibitor 2A/B; CIC, capicua transcriptional repressor; DA, diffuse astrocytoma; DMG, diffuse midlineglioma; EGFR, epidermal growth factor receptor; EPHB1, EPH receptor B1; FGFR1/3, fibroblast growth factor receptor 1/3; GBM, glioblastoma; HGG, high grade glioma; HGNT, high grade glioneuronal tumor; IDH1/2, isocitrate dehydrogenase 1/2; KDR, kinase insert domain receptor; LGG, low grade glioma; LGNT, low grade glioneuronal tumor; LP, lumber puncture; mut, mutant; MLH3, mutL homolog 3, MSH6, mutS homolog 6; N/A, not available, PDGFRA, platelet derived growth factor receptor alpha; PIK3CA, phosphatidylinositol-4,5-bisphosphate 3-kinase catalytic subunit alpha; PIK3R1, phosphoinositide-3-kinase regulatory subunit 1; PPM1D, protein phosphatase, Mg^2+^/Mn^2+^ dependent 1D; PTCH1, patched 1; PTEN, phosphatase and tensin homolog; pTERT, promoter of telomerase reverse transcriptase; OD, oligodendroglioma; VAD, ventricular access device (Ommaya reservoir); WGS, whole-genome sequencing; WES, whole-exome sequencing; wt, wild-type.

Wang et al. reported the earliest study of NGS analysis of cfDNA from CSF. The authors analyzed CSF-derived cfDNA from 35 primary brain tumors, including 29 gliomas, using targeted sequencing, followed by WES [81]. A total of 26 of the 35 patients (74%) had detectable levels of tumor DNA. In 20 cases of glioma, including low-grade gliomas, tumor-specific mutations were detected in 14 patients (70%). Pan et al. used a custom panel targeting 68 genes to detect tumor-specific alterations in CSF cfDNA from 57 cases of brainstem glioma in 82.5% of cases (47/57) [104]. When tumor-specific alterations were detected in the primary tumor, at least one matched alteration in CSF cfDNA was found in 97.3% (36/37), and all alterations were detected in 83.8% (31/37). Notably, tumor-specific alterations were readily detected in CSF cfDNA in 30% (3/10) of cases in which alterations were not detected in the tumor DNA. The initial barrier to NGS-based searches was low detection sensitivity; however, technological advancements are gradually overcoming this hurdle. The probability of detecting tumor-specific genetic alterations is now comparable to that of digital PCR in high-grade gliomas. The ability to track genetic alterations that reflect temporal and spatial heterogeneity is expected to provide a basis for precision oncology, enabling companion diagnoses through more detailed molecular background searches.

## 4. Clinical Practice and Molecular Marker

### 4.1. Isocitrate Dehydrogenase (IDH) 1/2

*IDH 1/2* is one of the most important and influential genes in gliomas [109]. IDH mutation includes a point mutation in the arginine at codon 132 of *IDH1* or the arginine at codon 172 of *IDH2*. IDH mutation occurs in the early stage of glioma development [139] and alters enzyme function to produce the oncometabolite 2-hydroxyglutaric acid (2-HG) [140], which is thought to cause glioma-specific methylation of CpG islands (G-CIMP), which is tumorigenic [141]. G-CIMP is a pivotal genome-wide methylation alteration in IDH-mutant gliomas, which may be stratified into two phenotypes according to their methylation level: G-CIMP-high and G-CIMP-low. Specifically, it has been suggested that the G-CIMP-low phenotype has poorer prognosis [122,142,143].

IDH status is a crucial marker for distinguishing glioblastoma, the glioma with the poorest prognosis, from low-grade gliomas, including the majority of astrocytomas and oligodendrogliomas. Digital PCR, an assay to detect SNVs with high sensitivity, is useful, and there have been many successful reports in the past [83,92,100]. NGS has also proven effective in detecting these SNVs [53,81,92,100,103,104,105,106]. Since IDH mutation is known to occur early in tumorigenesis and persists throughout [139], it is expected to be a useful marker in liquid biopsy from diagnosis to monitoring.

However, liquid biopsy in IDH-mutant gliomas tends to be less successful compared to GBM, owing to their less aggressive nature. This is particularly true for CSF obtained by lumbar puncture to diagnose, while liquid biopsy using intracranial CSF is more successful [83,92,100]. Moreover, in glioblastomas, IDH-wild-type tumor cells are indistinguishable from normal cells. Therefore, IDH mutation detection should be interpreted in combination with other positive findings, such as p*TERT*, *EGFR*, and Chr +7/−10.

### 4.2. Histone H3 Mutations

Mutations in histone H3 are important for the definitive diagnosis of IDH-wild-type gliomas, including pediatric-type midline gliomas. A missense mutation of H3 lysine 27 to methionine (H3 K27M) has been identified in DMG. This mutation occurs in H3.3 (encoded by *H3F3A*) or H3.1 (encoded by *HIST1 H3B/C*), and rarely in H3.2 (encoded by *HIST2 H3C*) [118,144]. Another mutation in H3F3A, G34R/V, in which glycine 34 is replaced by arginine or valine, has been identified as a cause of diffuse hemispheric glioma in adolescents and young adults [145]. H3 K27M alterations and G34R/V mutations are mutually exclusive [146,147].

DNA wraps around histone proteins to form chromatin; H3 is a core histone protein. H3 K27M mutation is located in the N-terminal histone tail; the K27M mutation reduces H3 with trimethylated lysine 27 (H3 K27me3), which is thought to cause epigenomic dysregulation and tumorigenesis [148,149]. A recent study demonstrated the efficacy of ONC201 for H3K27M-DMG. This study revealed that ONC201 disrupts integrated metabolic and epigenetic pathways, leading to a reversal of pathognomonic H3K27me3 reduction [93,102]. In the G34R/V mutation, it has been shown that it does not directly participate in the methylation of G34, but affects the methylation of the adjacent K36 (H3 K36me2/3) [150]; however, unlike K27M, the manner in which G34R/V exerts its dominant effect on H3 biology and results in tumorigenesis remains to be elucidated [151].

The hotspot region of the H3 mutation is another good target for detection using digital PCR, similar to the IDH mutations. Research has shown that the *H3F3A* K27M mutation can be detected in CSF obtained by lumbar puncture and successfully monitored by quantifying the mutation [93]. In particular DMGs benefit most from liquid biopsy because of their location, which makes biopsy risky or sometimes impossible, such as in the brainstem or deep in the brain. However, reports on liquid biopsy for *HIST1 H3B/C* and G34R/V are limited, partly because of their low prevalence [56,76,152]. In general, NGS assays may be suitable for the comprehensive evaluation of multiple H3 targets [73,104].

### 4.3. Telomerase Reverse Transcriptase Promoter (pTERT) Mutation

The ends of the chromosomes are protected by telomeres. Telomeres shorten with DNA replication, and DNA replication stops when the Hayflick limit is reached in normal somatic cells. *TERT* encodes a subunit of telomerase, and its expression is increased by mutations in one of the two hotspots in its promoter region in some gliomas [153,154]. p*TERT* mutation is observed in glioblastoma and oligodendroglioma, and p*TERT* mutation is the basis for the diagnosis of IDH-wild-type glioblastoma and IDH-mutant oligodendroglioma [2,153,155,156,157].

p*TERT* mutation is another SNV with known hotspots and can be detected by digital PCR. However, p*TERT* has GC-rich sequences (>80% GC rate) and its amplification requires overcoming technical challenges [99,158,159]. Currently, there are an increasing number of reports of successful detection using digital PCR, including in plasma [83,99,100]. In combination with IDH status, IDH-wild-type and p*TERT* mutants are classified as GBM, IDH-mutants, and p*TERT* wild-type as astrocytoma, IDH-mutant, and p*TERT* mutants as oligodendrogliomas [83,100]. However, for this concept to be valid, the false-negative rates for both IDH and p*TERT* mutations must be sufficiently low, particularly in lower-grade gliomas.

### 4.4. Epidermal Growth Factor Receptor (EGFR) Amplification and Chr +7/-10

DNA copy number and transcriptome analyses revealed epidermal growth factor receptor (*EGFR*) amplification and a combination of chromosome 7 gain and chromosome 10 loss (Chr +7/−10) as the characteristic molecular profiles of GBM [160,161,162]. The accumulation of prognostic analysis for clinical features and genome-wide methylome analysis has shown that IDH-wild-type gliomas with these copy number alterations have similar characteristics to glioblastomas, regardless of histological grade [8,121,163]. In addition to p*TERT* mutations, IDH-wild-type gliomas with these copy number alterations are classified as glioblastomas in the latest CNS tumor classification [2,156].

*EGFR* is a receptor tyrosine kinase and potent oncogene located on chromosome 7p. *EGFR* amplification primarily activates the *PI3K*/*Akt* and MAPK pathways and transmits downstream signals that promote cell proliferation. In GBM, the former pathway is particularly activated, and the phosphatase and tensin homolog (*PTEN*) on chromosome 10q, a tumor suppressor gene that suppresses this pathway, is deleted or mutated in glioblastomas [160,164,165].

Digital PCR is not suitable for copy number assessment of the entire locus, including *EGFR* and *PTEN*, or at the chromosomal level; previous research was limited to the detection of *EGFR* amplification [100]. NGS-based methods are effective for copy number analysis because of the assay characteristics [53,100,103]. Although the number of samples was limited, there is a report of successful copy number evaluation using MLPA [84], and it may be possible to evaluate copy numbers at a lower cost by using this assay.

### 4.5. Cyclin-Dependent Kinase Inhibitor 2A/B (CDKN2A/B) Homozygous Deletion

Cyclin-dependent kinase inhibitor A/B (*CDKN2A/B*) is a tumor suppressor gene in chromosome 9p, which was noted to be deleted in gliomas before the IDH mutation was discovered [166]. *CDKN2A* encodes p16INK4a and p14ARF and *CDKN2B* encodes p15INK4b. p16INK4A and p15INK4b induce G1 cell cycle arrest by inhibiting the activity of cyclin-dependent kinases 4 and 6 (*CDK4* and *CDK6*), which phosphorylate the RB protein, and p14ARF activates p53 by binding to MDM2 and promoting its rapid degradation [167,168]. Homozygous deletion of *CDKN2A/B* is known to be a common gene alteration in GBM [161,166], but subsequent analysis has shown that it is a characteristic of IDH-mutant gliomas, especially astrocytomas [155,169,170,171,172]. This provided the basis for assigning *CDKN2A/B* homozygous deletions of the same grade 4 as GBM in the latest brain tumor classification for IDH-mutant astrocytoma [2,173].

Therefore, confirming *CDKN2A/B* status is essential in gliomas with IDH-mutant and wild-type p*TERT*. Although digital PCR has been employed for liquid biopsy to detect these mutations [100], NGS-based assays are suitable for copy number evaluation, as previously described, and have been practically used [11,53,100]. As mentioned above, *CDKN2A* homozygous deletions have also been detected using MLPA, presenting an alternative method for this assay [84].

### 4.6. Chromosomes 1p/19q Codeletion

Co-deletion of chromosomes 1p and 19q was first reported in the same period as the deletion of *CDKN2A/B* [174,175] and has since become an essential molecular feature in the diagnosis of oligodendroglioma [2,176], following the discovery of IDH mutations. The 1p/19q codeletion was reported to be caused by the deletion of an unbalanced translocation t(1;19)(q10;p10), occurring near the centromeres of each chromosome [177,178]. Furthermore, NGS revealed the deletion of *FUBP1* at 1p and *CIC* at 19q in the remaining allele, suggesting that these genes are associated with tumorigenicity [179]. While 1p/19q codeletion is specific to oligodendroglioma, it is accompanied by loss of heterozygosity (LOH) of 1p or 19q in other gliomas, especially in GBM [180]. Therefore, for the diagnosis of oligodendroglioma, it is essential to extensively evaluate the entire 1p and 19q chromosomes and to confirm the presence of IDH-mutant.

In liquid biopsy, NGS assays capable of this comprehensive analysis are practical [53,103]. The presence of both IDH-mutant and pTERT mutations is useful as a surrogate marker for oligodendrogliomas [83,153]. As mentioned, in such cases, the combination of the two SNVs can be sensitively diagnosed through digital PCR.

## 5. Discussion and Future Perspectives

With recent advances in molecular biology and understanding of the molecular background of glioma, liquid biopsy in glioma continues to accumulate results. The goal of liquid biopsy has two major aspects: minimally invasive diagnosis in cases where surgery is not feasible and where real-time monitoring of treatment efficacy after initial treatment is required [34,35,36,37,38,39,40].

Gliomas are not uncommonly located in high-risk regions including the brainstem and eloquent areas. In such cases, it is important to make a reliable diagnosis using a minimally invasive method rather than radical surgical resection. The development of liquid biopsy has the potential to allow the characterization of entire tumors with high heterogeneity. In particular, CSF, which is rich in ctDNA, is an ideal resource. Liquid biopsy may allow for diagnosis based on the molecular background of the tumor and can be used as a basis for introducing initial treatment, including the selection of appropriate chemotherapy. Highly sensitive assays such as digital PCR are superior in detecting changes in a small number of target genes, while comprehensive assays such as NGS are more effective in detecting copy number alterations and gene mutations without hotspots. Copy number analysis is vital for the current classification of gliomas, and NGS is a promising methodology for liquid biopsies. In particular, it can detect copy number alterations and mutations at numerous loci that should be confirmed in pediatric-type gliomas and is also expected to detect fusion genes [181]. Furthermore, companion diagnostics may enable the realization of precision medicine, which links gene alterations to effective drugs.

Recent technological developments have led to the application of a new generation of sequencing techniques, as exemplified through Nanopore sequencing [82,182]. Recently, Afflerbach et al. reported a successful genome-wide methylome and copy number analysis of cfDNA in the CSF of glioma patients [182]. This new technology, which enables methylome analysis using extremely small amounts of DNA compared to the conventionally required DNA input, is noteworthy and may open new perspectives for liquid biopsy if it can achieve the inherently relatively high failure rate and improved sensitivity of Nanopore sequencing.

Real-time monitoring using liquid biopsy is a challenging task; however, it is an ideal goal. Except for low-grade circumscribed gliomas, gliomas are essentially infiltrating tumors [183,184,185,186]. The highest-grade glioblastoma is considered completely unresectable, and recurrence is inevitable. Therefore, it is necessary to monitor the course of the disease after initial treatment. Longitudinal monitoring of ctDNA after surgical resection may allow quantitative evaluation of residual disease activity. For example, if ctDNA is decreased through initial treatments such as chemotherapy and radiation, it may be assumed that the response to treatment is shown and the time to recurrence may be prolonged. In addition, the true state can be distinguished from pseudoprogression, pseudoresponse, and radiation necrosis, which can confound the evaluation of glioma recurrence. Indeed, other cancers have been evaluated in prospective studies of blood-based ctDNAs and have been shown to contribute to patient stratification [18,21,22]. It has been found that ctDNA detection is possible 7.9–11.0 months before relapse is clinically confirmed [18,19,20]. Furthermore, it has been reported that the appearance of treatment-resistant clones can be observed even before the clinical signs of recurrence [187,188]. In gliomas, serial evaluation of ctDNA using ddPCR has shown that this concept is realistic [93]. Similar to reports on other tumors, changes from baseline in VAF are suggested to be associated with treatment response and may be useful in predicting recurrence, since VAF increases prior to recurrence.

There are two major challenges in the liquid biopsy of gliomas. The first is the invasiveness of CSF sampling. Although lumbar puncture is less invasive than surgery, it is more invasive than peripheral blood sampling. Furthermore, lumbar puncture is contraindicated in patients receiving anticoagulant medications, as well as those with space-occupying lesions (due to increased intracranial pressures), posterior fossa masses, and/or coagulopathies [189]. Repeated lumbar punctures are burdensome to the patient due to the subsequent monitoring requirements, even if they are minimally invasive at the time of initial diagnosis. Alternatively, implantation of an Ommaya reservoir allows for more minimally invasive CSF collection when limited to postoperative monitoring [56]; however, the permissibility of placing a device in cases in which intermittent CSF evacuation or intrathecal drug administration is not planned is controversial. Therefore, CSF-based liquid biopsy methods have superior sensitivity and specificity for gliomas, a shift to peripheral blood-based methods is desirable if the aforementioned technical issues can be addressed in order to achieve comparable diagnostic performance. Another issue is the methodology used in the tests. Digital PCR is the most complete method with extremely high sensitivity; however, it can only evaluate known mutations. On the other hand, NGS can detect unanticipated genetic abnormalities but is generally less sensitive, often fails to identify driver mutations, and is expensive. Considering the characteristics of the methods, it may be feasible to monitor VAF using digital PCR targeting IDH1/2, pTERT, H3F3A, or other driver mutations identified via tumor tissue sequencing, and then evaluate for new mutations via NGS when events occur, such as recurrence This strategy may be realistic.

## 6. Conclusions

Interest in liquid biopsy for gliomas is growing rapidly, and evidence of its clinical value is accumulating annually. Although blood-based liquid biopsy is more difficult to perform in gliomas than in other types of cancers, gliomas have the unique advantage of using CSF as a promising resource. Currently, there are challenges in implementing CSF in clinical practice on a large scale; however, recent rapid technological advances and exploration of the molecular background of cancer, including gliomas, are expected to lead to the development of more sensitive assays and new methods for biomarker detection.

## Figures and Tables

**Figure 1 cancers-16-01009-f001:**
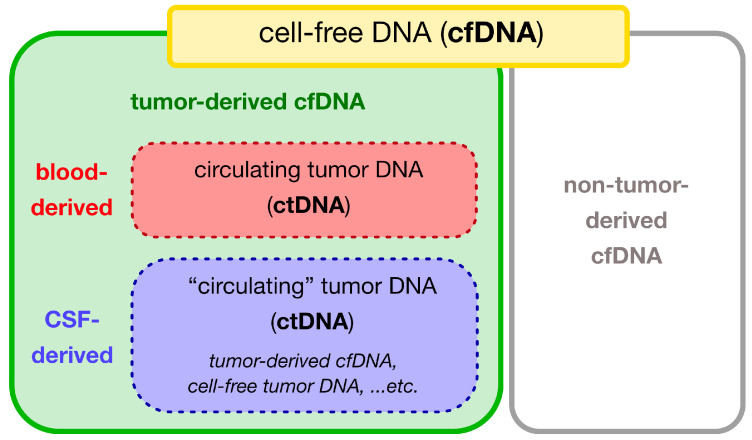
Venn diagram showing the inclusion relationship of terminology. Cell-free DNA (cfDNA) includes both normal cell- and tumor cell-derived DNA. Tumor-derived cfDNA can be specifically distinguished from circulating tumor DNA (ctDNA). Although the abbreviation ctDNA was originally intended for blood-derived cfDNA, it has also been used for cerebrospinal fluid-derived cfDNA.

## Data Availability

The data can be shared up on request.

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
