# Peer review of "Liquid Biopsy for Glioma Using Cell-Free DNA in Cerebrospinal Fluid"

_cancers, 2024, doi:10.3390/cancers16051009_

Round 1

Reviewer 1 Report

Comments and Suggestions for Authors

Dear Authors,

I enjoyed reading your review. Congratulations! Is is a useful update of current knowledge in the era of liquid biopsy biomarkers especially in tumors where classical biopsies are hard to perform. I have very few comments in the attached pdf.

Best of luck!

Author Response

Dear Reviewer 1,

We appreciate your review and comments. We appreciate your attention to detail; it is important to point out that "in vitro" is not italicized and the "latest" WHO classification of original Figure 1 as this writing is per the 2021 classification. In the revised manuscript, the term "in vitro" was used correctly. (However, Figure 1 was pointed out by another reviewer as unnecessary and was removed.)

Reviewer 2 Report

Comments and Suggestions for Authors

In the manuscript titled “Liquid biopsy for glioma using cell-free DNA in cerebrospinal fluid”, these authors provide an overview of the current status of CSF-based cfDNA-targeted liquid biopsy for gliomas. This review delineates distinctions from liquid biopsies for extracranial cancers, addresses prevailing challenges, and explores future prospects in this field.

Minor revision has to be done before this manuscript could be accepted for publication in the Cancers.

Minor comments:

1. On page 1, the author marked by the star sign is conflicted with the correspondence author.

2. In Figure 2, from the legend, the area of non-tumor-derived cfDNA (gray) and the tumor-derived cfDNA (green) should be independent rather than inclusive.

Comments on the Quality of English Language

The English language should be further polished by native speakers or proofreading services.

Author Response

Dear Reviewer 2,

Thank you for your peer review and comments. We are honored by your appreciation for the value of our manuscript.

1. The asterisk indicating the responsible author was incorrect and has been corrected accordingly.

2. We appreciate you pointing out the inclusion in Figure 2. In the original figure, as you pointed out, non-tumor-derived cfDNA includes the tumor-derived cfDNA. The two are opposites and not mutually inclusive. Thus, we have revised the figure (Figure 2 was renamed as Figure 1 because in response to feedback from another reviewer that the original Figure 1 was unnecessary and to be removed).

Reviewer 3 Report

Comments and Suggestions for Authors

I reviewed the paper from Ryosuke Otsuji et colleagues, and while the work doesn't seem poorly written and is intellectually stimulating, it undoubtedly requires an extensive manuscript revision to significantly reduce its volume and enhance readability. I am rather skeptical about some assumptions made in the introduction. I don't believe that the role of liquid biopsy will surpass the effectiveness and accuracy of tissue biopsy; instead, its advantage lies in minimally invasive procedures rather than sensitivity and specificity. Overall, the manuscript appears well-written and comprehensive, but it suffers from redundancy. Many paragraphs seem unnecessary, deviating from the article's context and content. For instance, Image 1 on tumor classification is entirely redundant, assuming that readers already possess this knowledge or can retrieve it from appropriate citations without losing information. Additionally, I recommend incorporating citations for recent reviews on the tumor microenvironment and its therapeutic implications in gliomas in the introductory section (PMID: 37371615 and PMID: 35456984).

Moreover, I propose significant reductions in Sections 3.1 and 3.2, particularly eliminating historical and technical details about various methodologies. These details are redundant, making the reading challenging and dispersing information. The same applies to Sections 4.1, 4.2, etc. For example, in Section 4.1, I would retain only the last sentence, as authors should focus not on specific techniques or mutations but on how these are highlighted in cell-free liquid biopsy.

Author Response

Dear Reviewer 3,

Thank you for your review and useful comments. 

・As you pointed out, liquid biopsy is not a tissue biopsy superset. Specimens should be obtained by surgery for definitive diagnosis as far as possible. The previous version of the manuscript described the disadvantages of surgery, which were emphasized more than necessary, thus we have revised the wording to be more neutral. Potential sampling bias of tissue biopsy (especially in cases where only a small amount of tissue can be obtained) and longitudinal monitoring due to the minimally invasive nature of the procedure represent important backgrounds that are currently driving research on liquid biopsy. The wording has been selected to clarify that liquid biopsy is not a replacement for surgery but a complement to it.

"... in some cases, particularly in the deep brain or brainstem region, it is even riskier or impossible to perform a biopsy due to its location. In such cases, minimally invasive diagnostic options such as liquid biopsy, may be beneficial in conjunction with surgery. "

・In accordance with your suggestion, we have reorganized the text. In particular, we made major deletions to section 3 and removed image 1 (we attempted to reorganize sections 4.1 and 4.2 as well; however, reviewer 4 requested a description of G-CIMP and inclusion of information on the ONC201 drug. Therefore, we could not reduce the description of the molecular background of each gene. Subsequently, sections 4.3 and beyond were preserved with a minimum description of the molecular background to maintain balance with sections 4.1 and 4.2)

"3.1. Digital PCR...This highly sensitive technique successfully detected ctDNA targeting H3 K27M and pTERT mutations in gliomas [56,76,83,98,99]. Recently, Muralidharan et al. utilized a novel ddPCR assay that focused on the two most common pTERT mutations and demonstrated improved sensitivity and specificity for plasma ctDNA detection and the long-term monitoring of gliomas [95]. In addition, Fontanilles et al. reported the monitoring of plasma ctDNA and detection of pTERT mutations in the NCT02617745 study [84]. However, pTERT mutations were detected in only 2/45 (4.4%) of mutation-positive cases, and the two detected cases were histologically gliosarcoma [84]. Therefore, even with the high sensitivity of ddPCR, it is difficult to evaluate and monitor the VAF of pTERT mutations in the plasma ctDNA. As mentioned above, it is practical to use CSF, which is rich in ctDNA. Izquierdo et al. evaluated ctDNA in the plasma, serum, and CSF using ddPCR in 32 pediatric patients with HGG and DMG [72]. In their study, ctDNA was most frequently detected in CSF (6/9, 66.7%), whereas only 26% (7/27) and 33% (2/6) were detected in plasma and serum, respectively [72]. Table 1 summarizes the representative studies utilizing digital PCR on CSF and shows that digital PCR can detect variant alleles with high sensitivity of 70–100% in glioblastoma and 80–100% in DMG [13,56,76,83,92,93,100]. "

"... In a study using an in vitro model in which H3F3A K27M-mutant DMG cells were co-cultured with normal human astrocytes, the presence of H3F3A K27M mutation in cfDNA in the culture medium was evaluated using ddPCR, and H3F3A K27M mutant ctDNA was shown to reflect tumor growth [101]. Despite frequent culture medium replacement to mimic the constant production and absorption of cerebrospinal fluid, an increase in cfDNA with the H3K27M mutation in the culture medium was observed in proportion to cell proliferation. Furthermore, irradiation to the tumor cells resulted in a dramatic increase in cfDNA for approximately 72–120 hours, followed by a gradual decrease. These results suggest that ddPCR may be useful for monitoring ctDNA in response to treatment. In the ONC201 trial (NCT03416530), ..."

"3.2. Next-generation sequencing (NGS)...Applications of NGS include targeted gene panels [110,111], whole-exome sequencing (WES) [112–115], and whole-genome sequencing (WGS). WES captures protein-coding exons that span 1–2% of the entire human genome. WES sequences 70–100× or more "depth" (coverage of target regions) for each sample. Tumor tissue-based cancer genome sequencing has been performed previously, mainly using WES. Currently, a vast amount of mutation data on protein-coding regions related to cancer has been accumulated, and most driver genes found at high frequencies have been identified [115, 116]. Somatic mutation data accumulated by pan-cancer exome analysis were compiled in the Catalogue of Somatic Mutations in Cancer (COSMIC) and released in the public domain [111]. Currently, Furthermore, cancer genome analysis of tumor tissues can be further extended to WGS, which covers the entire genome, including non-coding regions such as promoters and regulatory regions, and can also detect variants in the genome structure, including translocations, inversions, and gene fusions. Such genome-wide exploration This comprehensive exploration of the genome may lead to the discovery of new and currently undiscovered oncogenic aberrations [117–125]."

"...Highly sensitive NGS-based assays have been shown to allow for detection of ctDNA in plasma samples from patients with glioma [52,54,72,136,137]; however, For example, the Gardant360 NGS assay targets 512 exons across 54 genes. Piccioni et al. applied this assay to 419 primary brain tumor patients and detected ctDNA positivity in the plasma in approximately 50% of primary brain tumors and 55% of GBM [50]. Using the same NGS platform, Bagley et al. found elevated plasma ctDNA levels in glioma patients and demonstrated a possible correlation between this quantitative increase in ctDNA, tumor burden, and progression-free survival [68,131]. Studies using proprietary sequencing assays have reported higher detection rates and tumor fractions [48,50,132]. However, in addition to biological and technical noise, genomic alterations associated with cancer have been found in plasma, even in that of healthy individuals, potentially confounding the interpretation [134,137,138]. "

"...Pan et al. utilized a custom panel targeting 68 genes to detect tumor-specific alterations in CSF cfDNA from 57 cases of brainstem glioma and achieved a detection rate of 82.5% (47/57) [104]. When tumor-specific alterations were detected in the primary tumor, at least one matched alteration in CSF cfDNA was found in 97.3% (36/37) of cases, with all alterations were detected in 83.8% (31/37) of cases. Notably, tumor-specific alterations were readily detected in CSF cfDNA in 30% (3/10) of cases where alterations were not detected in the tumor DNA. Miller et al. used MSK-IMPACT to analyze CSF cfDNA from 85 patients with glioma after therapeutic interventions including surgery, radiation therapy, and chemotherapy [49]. cfDNA-identified mutations showed a real-time genomic profile of the tumor. For example, comparing CSF and corresponding tumor tissue mutations, the 30 CSF-tumor pairs without hypermutation had a median shared mutation rate of 81.7%, whereas the six hypermutant pairs had a much lower frequency of shared mutations with a median rate of 19.6%. Conversely, in the six patients who underwent lumbar puncture and VP shunt within 3 weeks and in the five patients who underwent biopsy within 3 weeks after lumbar puncture, the genomic profiles of CSF and tumor DNA were identical, regardless of the presence or absence of hypermutations."

 ãƒ»Furthermore, we acknowledge that this review is focused on liquid biopsy using CSF cfDNA. However, presenting too much information may overwhelm readers and detract from conveying a clear and impactful message. We have included the following sentence in the introduction as a preview of the topics that will be touched upon in the upcoming sections.

"Specifically, Section 2 provides an overview of cfDNA/ctDNA-based liquid biopsy techniques and Section 3 summarizes existing methodologies. Section 4 summarizes the molecular background of various liquid biopsy targets for glioma and describes potential applications of these assays."

 ãƒ»A description of the tumor microenvironment and immunotherapy has been included in the introduction. We have also cited the two reviews you recommended (Citation 31 and 32).

Furthermore, advances in our understanding of the tumor microenvironment of glioma and advances in immunotherapy have underscored the importance of assessing bodily fluid biomarkers for therapeutic planning and clinical management [30–33]"

Reviewer 4 Report

Comments and Suggestions for Authors

The review focuses on the potential of CSF-based cell-free DNA (cfDNA) liquid biopsy in glioma diagnosis. It emphasizes the challenges of traditional biopsy methods, especially when tumors are deeply located. The review highlights the promising role of cfDNA in CSF, considering the limitations of blood-based cfDNA analysis due to the blood-brain barrier. The use of digital PCR and next-generation sequencing in detecting tumor-specific genetic alterations in CSF cfDNA is explored, discussing how this differs from liquid biopsies in extracranial cancers.

 Minor comments:

1.       The review would benefit from the inclusion of the ONC201 trial (DOI: 10.1158/2159-8290.CD-23-0131). ONC201 is a significant advancement in treating certain advanced cancers, particularly brain tumors with the H3 K27M mutation. Its inclusion would add depth to the review.

2.       The review should discuss the limitations and challenges of using CSF for liquid biopsy compared to blood, despite the advantages in the context of CNS tumors.

3.       Expand on the clinical implications of using cfDNA in CSF as a non-invasive biomarker for glioma diagnosis, especially compared to blood-based methods.

4.       A detailed exploration of G-CIMP low and high profiles in gliomas would be valuable, as these profiles have significant diagnostic and prognostic implications.

Author Response

Dear Reviewer 4,

We appreciate your review and useful comments. We are honored by your deep understanding of the intent of our manuscript.

1. We appreciate you pointing out the issue with respect to ONC201. As you mentioned, the H3K27M mutant ctDNA evaluated with ddPCR used in ONC201 reflects tumor activities and is recognized as an ideal model for liquid biopsy (reference 93 and explained in the section 3.1). However, the therapeutic outcome of ONC201 drugs is insufficiently reported. We added clinical results obtained from the trial (reference 102) and mentioned the possibility of effective drugs for H3K27M mutant DMG using the sentence shown below:

"A recent study demonstrated the efficacy of ONC201 for H3K27M-DMG. This study revealed that ONC201 disrupts integrated metabolic and epigenetic pathways, leading to a reversal of pathognomonic H3K27me3 reduction [93,102]."

2. We added a note on the disadvantages of using CSF in section 5). Specifically, we have highlighted the limitation of lumbar puncture as a method to collect CSF. Moreover, it is evident that repeated lumbar punctures are more invasive than peripheral blood sampling.

"Furthermore, lumbar puncture is contraindicated in patients receiving anticoagulant medications, as well as those with space-occupying lesions (due to increased intracranial pressures), posterior fossa masses, and/or coagulopathies [189]. Repeated lumbar punctures are burdensome to the patient due to the subsequent monitoring requirements, even if they are minimally invasive at the time of initial diagnosis."

3. The difference in clinical application between blood-based and CSF-based liquid biopsy is due to their reliability (sensitivity and specificity). Especially in glioma, the use of plasma-derived cfDNA poses many challenges, and CSF is currently superior for diagnosis. Conversely, if technological advances are made in the detection of peripheral blood-derived ctDNA, liquid biopsy using blood-derived ctDNA will be superior, as it is for other cancer types. In section 5, we added the following sentences:

"Therefore, CSF-based liquid biopsy methods have superior sensitivity and specificity for gliomas, a shift to peripheral blood-based methods is desirable if the aforementioned technical issues can be addressed in order to achieve comparable diagnostic performance. "

4. We agree with your point that G-CIMP-high/low is an important factor that significantly affects prognosis. We have added the following sentences in section 4.1:

"G-CIMP is a pivotal genome-wide methylation alteration in IDH-mutant gliomas, which may be stratified into two phenotypes according to their methylation level; G-CIMP-high and G-CIMP-low. Specifically, it has been suggested that the G-CIMP-low phenotype has poorer prognosis [122,142,143]."

Round 2

Reviewer 3 Report

Comments and Suggestions for Authors

The requested changes have been made, and I see no obstacles to the publication of the manuscript in this form